# Involvement of the Iron-Regulated Loci *hts* and *fhuC* in Biofilm Formation and Survival of *Staphylococcus epidermidis* within the Host

Fernando Oliveira,[a,b,c] Tânia Lima,[c] Alexandra Correia,[c] Ana Margarida Silva,[c] Cristina Soares,[d] Simone Morais,[d] Samira Weißelberg,[b] Manuel Vilanova,[c,e,f] Holger Rohde,[b] Nuno Cerca[a]

aCentre of Biological Engineering, LIBRO – Laboratory of Research in Biofilms Rosário Oliveira, University of Minho, Braga, Portugal
bInstitut für Medizinische Mikrobiologie, Virologie und Hygiene, Universitätsklinikum Hamburg-Eppendorf, Hamburg, Germany
ci3S – Instituto de Investigação e Inovação em Saúde, Universidade do Porto, Porto, Portugal
dREQUIMTE-LAQV, Instituto Superior de Engenharia do Porto, Instituto Politécnico do Porto, Porto, Portugal
eIBMC, Instituto de Biologia Molecular e Celular, Universidade do Porto, Porto, Portugal
fICBAS-UP, Instituto de Ciências Biomédicas de Abel Salazar, Universidade do Porto, Porto, Portugal

**ABSTRACT** *Staphylococcus epidermidis* is a major nosocomial pathogen with a remarkable ability to persist on indwelling medical devices through biofilm formation. Nevertheless, it remains intriguing how this process is efficiently achieved under the host's harsh conditions, where the availability of nutrients, such as essential metals, is scarce. Following our previous identification of two iron-regulated loci putatively involved in iron transport, *hts* and *fhuC*, we assessed here their individual contribution to both bacterial physiology and interaction with host immune cells. Single deletions of the *hts* and *fhuC* loci led to marked changes in the cell iron content, which were partly detrimental for planktonic growth and strongly affected biofilm formation under iron-restricted conditions. Deletion of each of these two loci did not lead to major changes in *S. epidermidis* survival within human macrophages or in an *ex vivo* human blood model of bloodstream infection. However, the lack of either *hts* or *fhuC* loci significantly impaired bacterial survival *in vivo* in a murine model of bacteremia. Collectively, this study establishes, for the first time, the pivotal role of the iron-regulated loci *hts* and *fhuC* in *S. epidermidis* biofilm formation and survival within the host, providing relevant information for the development of new targeted therapeutics against this pathogen.

**IMPORTANCE** *Staphylococcus epidermidis* is one of the most important nosocomial pathogens and a major cause of central line-associated bloodstream infections. Once in the bloodstream, this bacterium must surpass severe iron restriction in order to survive and establish infection. Surprisingly, very little is known about the iron acquisition mechanisms in this species. This study represents the first report on the involvement of the *S. epidermidis* iron-regulated loci *hts* and *fhuC* in biofilm formation under host relevant conditions and, most importantly, in survival within the host. Ultimately, these findings highlight iron acquisition and these loci in particular, as potential targets for future therapeutic strategies against biofilm-associated *S. epidermidis* infections.

**KEYWORDS** biofilms, innate immunity, iron

Healthcare-associated infections have become a major threat to public health worldwide (1). *Staphylococcus epidermidis* has emerged as a major cause of nosocomial infections and belongs to the most common etiologic agents of infections associated with the use of indwelling medical devices and implants, particularly catheters

Address correspondence to Nuno Cerca, nunocerca@ceb.uminho.pt.

The authors declare no conflict of interest.

and prosthetic joints (2). Furthermore, there has been a global spread of multidrug-resistant *S. epidermidis* lineages (3), which has led to an increased shortage of therapeutic options for these kinds of infections.

As a major cause of device-associated bloodstream infections (4), *S. epidermidis* must have the ability to thrive in the severely iron-restricted bloodstream (5). Surprisingly, very little is known about the iron acquisition mechanisms of this species, which might be attributable, at least in part, to the inherent challenges of genetically manipulating staphylococci, particularly *S. epidermidis* (6). Iron acquisition has been suggested as a key process in bacterial pathogenicity, since pathogens face severe restriction in the availability of this metal, a phenomenon usually referred to as "nutritional immunity" (7). To surpass iron limitation, bacteria must have dedicated mechanisms to acquire this important metal and use it for their own cellular functions (8). One common strategy employed by bacteria to meet their iron requirements is to express specific surface receptors coupled to specialized transport systems that allow the translocation of iron across the cytoplasmic membrane. These include ATP-binding cassette (ABC) transporters, which mediate the import of iron, either in its ionic form or coupled to host-derived (e.g., transferrin and heme) (9) or bacteria-derived molecules (e.g., siderophores) (10). The study of iron acquisition mechanisms in staphylococci has been mostly focused on *S. aureus* (11–17), which has led to the identification and characterization of two siderophores belonging to the carboxylate family, staphyloferrin A (13, 14) and staphyloferrin B (15, 17), as well as their cognate transporters HtsABC and SirABC, respectively (11, 18).

The actual contribution of iron acquisition to *S. epidermidis* virulence has not been established to date. In this study, we establish the involvement of two recently identified iron-regulated loci (*hts* and *fhuC*, which together are predicted to encode an iron ABC transporter system) in *S. epidermidis* virulence, as demonstrated by their contribution to biofilm formation and survival within the host.

## RESULTS

**FhuC is critical for *S. epidermidis* to acquire iron.** We have previously identified two iron-regulated loci encoding components of an ABC transporter putatively involved in iron transport (19), which are well conserved across different *S. epidermidis* strains (Fig. S1). The B4U56_03565-03575 locus (henceforth called *htsABC*) is located immediately adjacent to the *sfaABCD* cluster, a putative siderophore biosynthetic pathway, and encodes two membrane permeases and one substrate-binding protein (Fig. 1a). Interestingly, this locus lacks the gene encoding the ATP-binding protein of a classical ABC transporter, something that has also been reported in *S. aureus* (13). Based on our previous identification of different iron-regulated genes (19), we hypothesized that the B4U56_10485 locus (henceforth called *fhuC*) encodes the missing protein (Fig. 1b). To confirm the involvement of these loci in iron acquisition, we deleted each locus in *S. epidermidis* 1457 following an allelic replacement strategy. Surprisingly, while a remarkable reduction in the iron cell content of the Δ*fhuC* strain was detected, deletion of *hts* led to a minor increase, although not statistically significant, in the iron cell content (Fig. 1c). Additionally, deletion of either *hts* or *fhuC* resulted in a growth defect under iron-restricted conditions, which was more evident between 6 and 8 h of growth. Nevertheless, both mutants were able to replicate under these conditions, achieving a cell density equivalent to the wild-type (wt) strain after 24 h (Fig. 1d). Since deletion of *hts* led to a marked rise in the cell iron content, we next studied the interplay among different iron acquisition systems at a transcriptional level. While siderophore overproduction by the Δ*hts* mutant was not apparent (Fig. S2), transcription analysis by quantitative real-time PCR (qPCR) of cells grown under iron-restricted conditions demonstrated that the mRNA levels of *sfaB* (putative siderophore synthetase) were significantly increased in this strain (Fig. 1e). This gene belongs to an iron-regulated locus (*sfaABCD*) that we recently characterized and demonstrated to play an important role in iron acquisition and bacterial survival within the

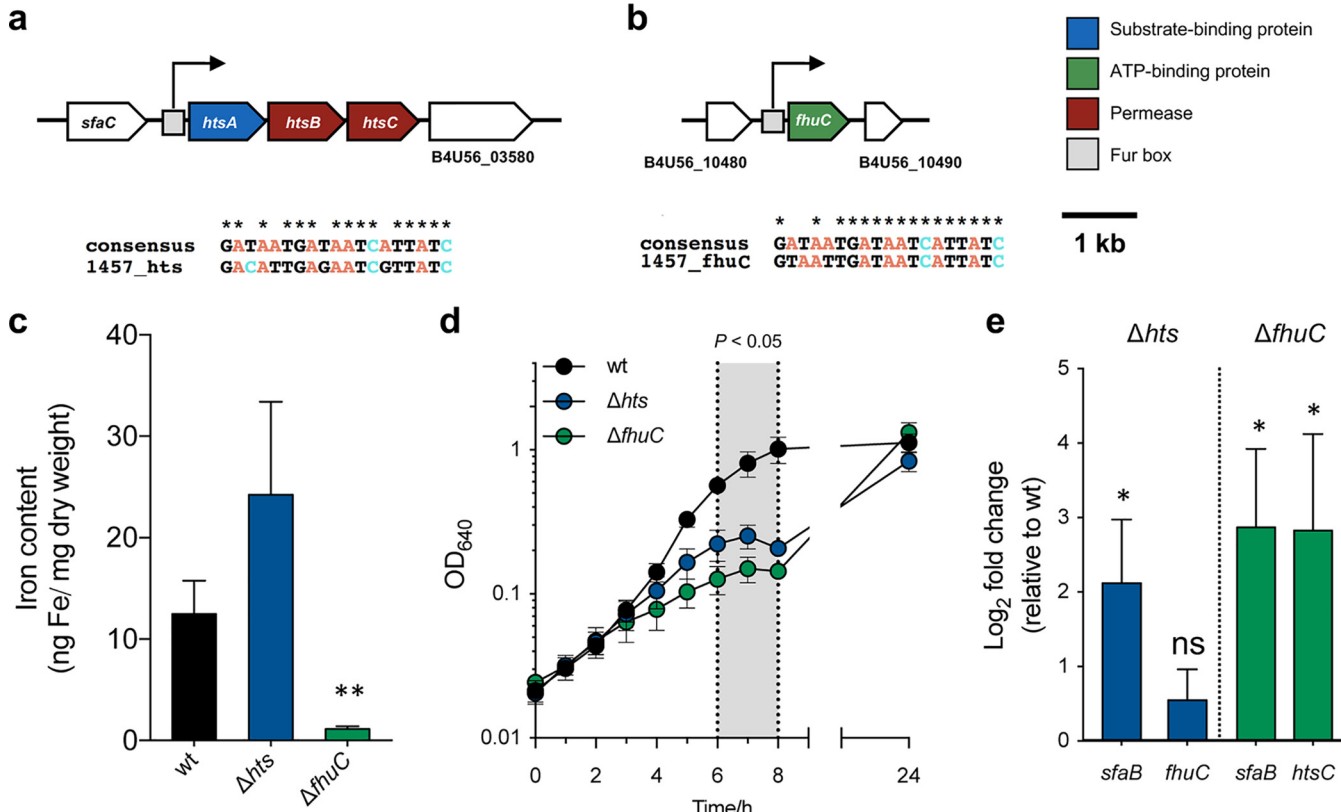

**FIG 1** Deletion of *hts* or *fhuC* has different outcomes in the iron cell content but not in the growth of *S. epidermidis* under iron-restricted conditions. (a, b) Genomic organization of the *htsABC* (a) and *fhuC* (b) loci in *S. epidermidis*, which encode putative components of iron ABC transporters. Assignments are based on the annotated genomes of *S. epidermidis* 1457 (accession number: CP020463). Open reading frames are indicated by arrows, which show the direction of transcription. Predicted transcriptional start sites are indicated by bent arrows. Putative Fur boxes are shown as gray boxes, and alignments with a consensus sequence are provided below. (c) Analysis of the cellular iron content by atomic absorption spectroscopy ($N = 4$). (d) Strains were allowed to grow for 24 h at 37°C, 120 rpm in CDM(Fe−), and growth was monitored as means of the optical density at 640 nm ($OD_{640}$) ($N = 3$ to 4). (e) Transcription of iron acquisition-associated genes after culture in CDM(Fe−) for 24 h. Fold change data were calculated according to Pfaffl method and log transformed ($Log_2$). Values above and below 0 indicate up- and downregulation of transcription, respectively, relative to wild type (wt) ($N = 3$). (c to e) Two-tailed *t* test (c) or two-way analysis of variance (ANOVA) (d, e). The bars (c, e) and symbols (d) represent the mean of biological replicates and the error bars represent the standard error of the mean (SEM). ns, not significant; *, $P < 0.05$ versus wt; **, $P < 0.01$ versus wt.

host (20). Additionally, a slight but not significant increase in *fhuC* mRNA levels was detected in this mutant (Fig. 1e). On the other hand, Δ*fhuC* cells exhibited significant higher mRNA levels of both *sfaB* and *htsC* genes relative to wt (Fig. 1e). By providing the missing loci in *trans* (p*hts* and p*fhuC*), the aforementioned effects could be fully/partly reversed (complementation with expression of *fhuC* in *trans* did not reverse the growth defect exhibited by Δ*fhuC*; Fig. S3a to c), strongly suggesting that the differences observed are attributable to the deletion of these loci.

Collectively, while these findings are compatible with a transcriptional compensation in response to disruption of specific iron acquisition pathways, it is worth mentioning that the effect on the iron content of these two mutants was strikingly distinct. Overall, our observations suggest a strong transcriptional interplay among different iron acquisition systems, with *fhuC* playing a more prominent role in the maintenance of iron homeostasis in *S. epidermidis*.

***S. epidermidis* relies on iron acquisition through the Hts-FhuC transporter to form biofilms.** To assess the contribution of iron acquisition to biofilm formation by *S. epidermidis*, recognized as its major virulence factor, we tested our mutant strains for their ability to form biofilms under different iron availability conditions. When grown in the iron-rich culture medium TSB (Fig. 2a), none of the mutant strains exhibited impaired biofilm formation ability. However, deletion of either *hts* or *fhuC* loci proved to be detrimental for biofilm formation under conditions of iron restriction (Fig. 2b). This effect could be partially or fully reversed either by providing the missing loci in

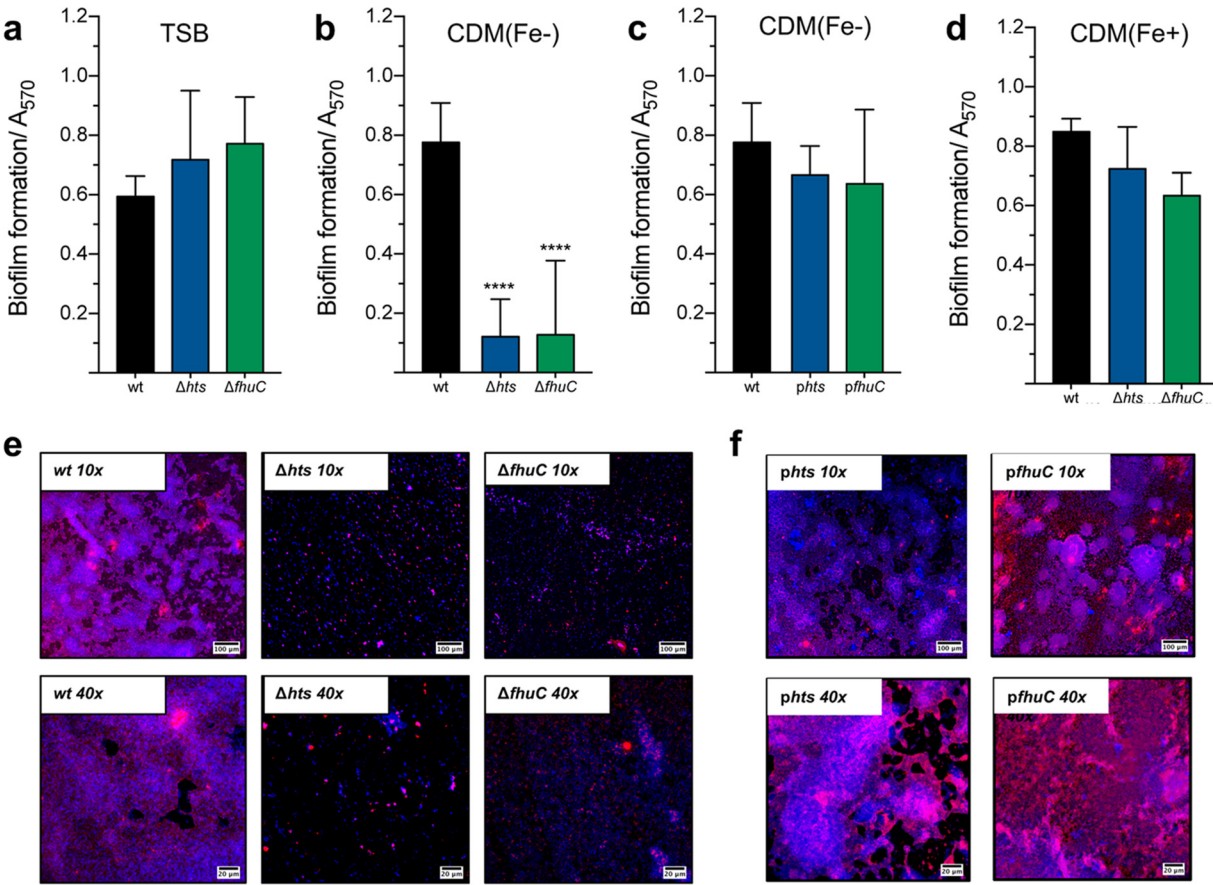

**FIG 2** Iron acquisition through the Hts-FhuC transporter is key for biofilm formation in low-iron environments. (a to c) The biofilm formation ability of each strain was evaluated in TSB (a), iron-restricted chemically modified medium (CDM[Fe−]) (b, c), and iron-enriched chemically modified medium (CDM[Fe+]) (c). The cells were allowed to grow statically for 24 h at 37°C on 96-well microplates. Biofilm quantification was performed through crystal violet staining ($N$ = 3 to 5). (c) Confocal laser scanning microscopy (CLSM) analysis of biofilms formed under iron-restricted conditions (CDM[Fe−]). Biofilms were allowed to grow on an 8-well chamber slide system in CDM(Fe−) at 37°C for 24 h. CLSM was used for biofilm structure analysis and PIA/PNAG production after appropriate staining with DAPI (4′,6-diamidino-2-phenylindole) (depicted in blue) and WGA-Texas Red (depicted in red). Representative images of Z-stack projections from two independent experiments are shown. Bar = 100 $\mu$m for 10× and 20 $\mu$m for 40×. ****, $P < 0.0001$ versus wt. (a to d) One-way ANOVA with Dunnett's multiple-comparison test. The bars represent the mean of biological replicates, and the error bars represent the standard deviation.

*trans* (Fig. 2c) or by supplementing the iron-restricted medium with iron chloride (Fig. 2d). Biofilms formed under iron-restricted conditions were further examined through confocal microscopy for the assessment of biofilm organization and the polysaccharide intercellular adhesin (PIA), also known as poly-N-acetyl-$\beta$-(1-6)-glucosamine (PIA/PNAG) content in the biofilm matrix (Fig. 2e). While wt formed thick biofilms containing a high density of cells widespread across the surface and large amounts of PIA/PNAG, $\Delta hts$ and $\Delta fhuC$ formed sparse biofilms mostly composed of cell clusters embedded in small amounts of PIA/PNAG. Gene complementation reversed the biofilm phenotype of all mutant strains (Fig. 2f). Taken together, our data reveal that *S. epidermidis* relies on the Hts-FhuC transporter to meet the iron requirements for biofilm formation when the availability of this metal is scarce.

**The lack of the Hts-FhuC transporter does not change survival of *S. epidermidis* inside human macrophages.** Macrophages are crucial players in the innate immune response against staphylococcal infections, as they phagocytose invading bacteria and expose them to a plethora of antimicrobial compounds (21). In addition, these cells are at the crossroads of innate immunity and iron metabolism by regulating iron homeostasis in response to infection (22). While biofilm formation has been shown to protect *S. epidermidis* from phagocytosis (23) and the action of antimicrobial peptides (24), the fate of *S. epidermidis* cells once phagocytosed is poorly understood. Our previous

findings on the importance of iron availability for *S. epidermidis* growth and biofilm formation (19) led us to hypothesize that the absence of specific iron acquisition systems may affect the survival of this bacterium inside phagocytic cells. To that end, we first studied the ability of *S. epidermidis* to survive within macrophages using the murine cell line RAW264.7 and a gentamicin protection assay (Fig. 3a). While these macrophages were able to control intracellular replication up to 12 h of infection, some level of bacterial proliferation became apparent at 24 h postinfection, although not statistically significant (Fig. 3b). Even though previous studies have reported intracellular persistence of *S. epidermidis* (25–27), replication within macrophages was an unprecedented finding. To further assess whether *S. epidermidis* could grow intracellularly in macrophages, we replicated our macrophage infection assays using human monocyte-derived macrophages (hMDMs) with different polarization phenotypes (M1- and M2-like macrophages). Unlike RAW264.7 cells, hMDMs cleared phagocytosed *S. epidermidis*, with no CFU detected in both M1- and M2-like hMDM cultures at 24 h postinfection. Of note, no significant differences were found in the number of bacterial cells residing intracellularly at 2 h postinfection relative to wt, with the exception of a numerical increase in Δ*fhuC* cells in M2-like macrophages (Fig. 3c).

**Hts/FhuC transporter is required for *S. epidermidis* survival within the host.** As *S. epidermidis* is a major pathogen causing bloodstream infections originating from indwelling medical device contamination (4), we hypothesized that iron acquisition provides a fitness advantage to this bacterium upon exposure to human blood. However, no significant changes relative to wt strain were registered, at both 2 and 4 h after exposure to human blood (Fig. S4). Nevertheless, when using a murine *in vivo* model of *S. epidermidis* bacteremia, we detected that mouse groups infected with either Δ*hts* and Δ*fhuC* mutants exhibited significantly reduced bacterial loads in blood, liver, and kidneys at 6 h postinfection compared with the wt strain (Fig. 4). In addition, Δ*fhuC*-infected mice also displayed a significantly reduced bacterial burden in the spleen. Conclusively, our *in vivo* findings provide significant evidence for the involvement of Hts-FhuC-mediated iron acquisition in *S. epidermidis* survival within the host.

## DISCUSSION

Unlike several other pathogens, the mechanisms behind iron acquisition in *S. epidermidis* remain largely unknown. Even though iron acquisition has been explored in more detail in the closely related species *S. aureus*, we have identified key differences between these two staphylococcal species that makes it difficult to directly translate findings from one species to another. First, *S. aureus* produces two different siderophores (staphyloferrins A and B) and expresses two transporters fully dedicated to their uptake (HtsABC and SirABC) (13, 14), whereas *S. epidermidis* has a single siderophore biosynthetic locus (19, 20). Second, genomic data available to date indicate that the *S. aureus* iron-regulated surface determinant (Isd), a dedicated system for acquisition of heme-bound iron (28, 29), is absent in *S. epidermidis*. Therefore, *S. epidermidis* seems to employ less complex mechanisms for iron acquisition, which may increase the relevance of each individual acquisition system for key virulence processes. This study represents a first approach to this hypothesis, with our focus being directed toward the iron-regulated loci *hts* and *fhuC*, which are predicted to encode the different components of an ABC transporter.

Specifically, we demonstrated the involvement of these loci in iron acquisition, as deletion of either *hts* or *fhuC* loci had a noticeable impact in the cell iron content (Δ*hts* = high iron cell content; Δ*fhuC* = low iron cell content). Analysis of transcript levels of iron acquisition-related genes revealed an upregulation of *sfaB* (putative siderophore synthetase) in both mutants when cultured under iron restriction. Furthermore, we detected a significant increase in *htsC* transcript levels in Δ*fhuC* mutant cells. However, none of these transcriptional changes were sufficient to restore the iron cell content of Δ*fhuC* mutant to wt levels, which indicates that the ATPase FhuC plays a central role in iron acquisition by *S. epidermidis*. Moreover, our results support some degree of redundancy

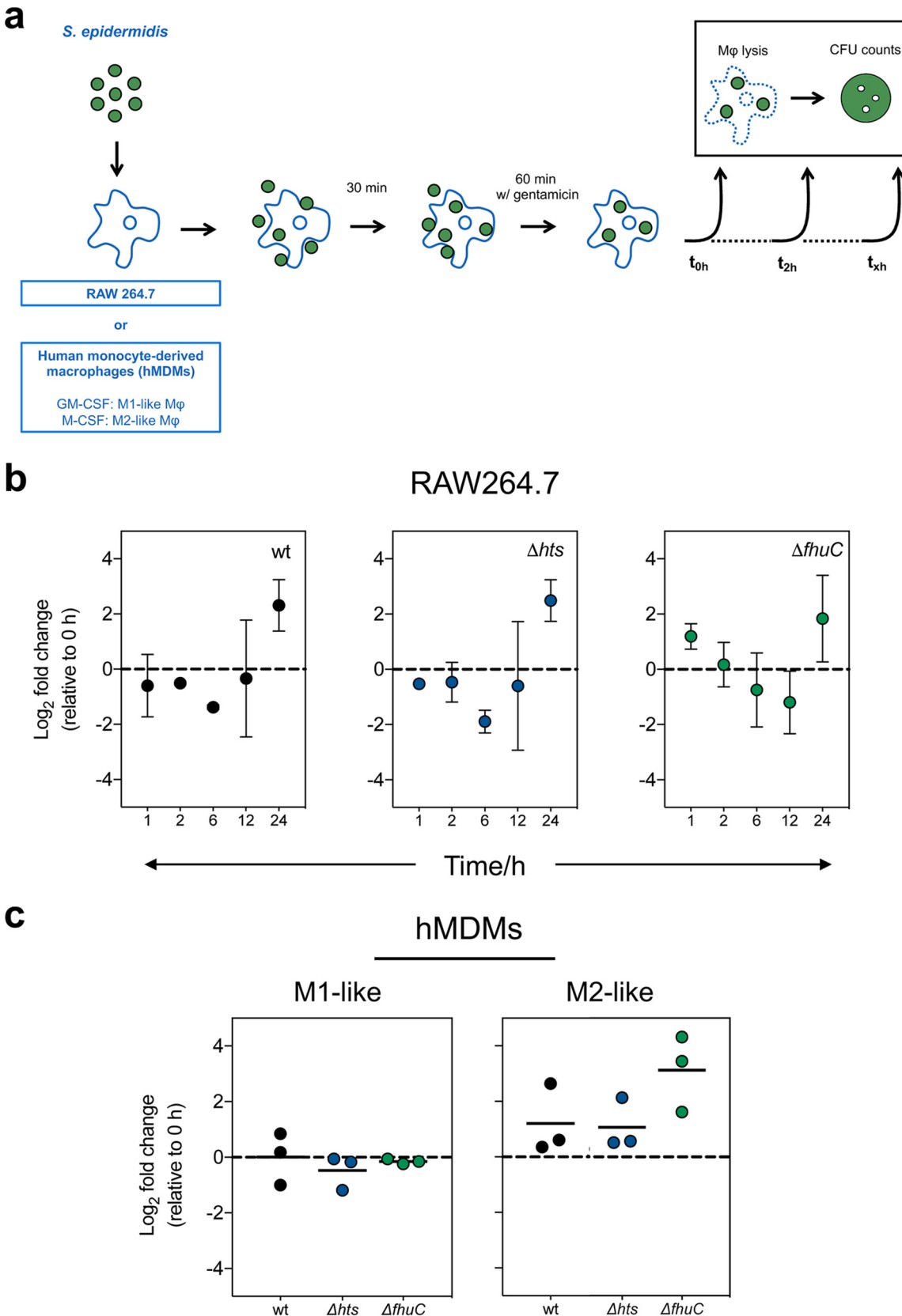

**FIG 3** The iron-regulated loci *hts* and *fhuC* do not contribute to the interaction between *S. epidermidis* and phagocytic cells. (a) Schematic diagram of the gentamicin protection assays used in this study. Macrophages (murine RAW264.7 or human monocyte-derived macrophages [hMDMs]) were infected with *S. epidermidis* as described in Materials and Methods. Bacterial cells were recovered from the

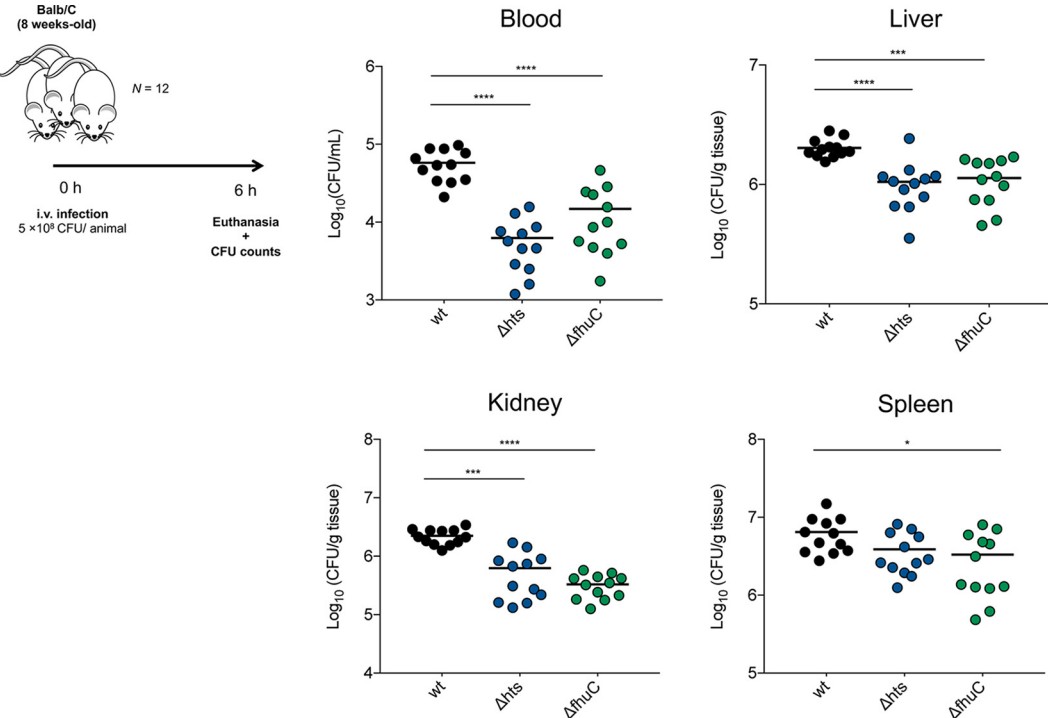

**FIG 4** *S. epidermidis hts* and *fhuC* mutants exhibit impaired survival within the host. BALB/c mice were infected intravenously with $5 \times 10^8$ CFU of *S. epidermidis* 1457 or its isogenic iron uptake (Δ*hts* and Δ*fhuC*) mutants. Six hours after infection, the mice were euthanized; blood, liver, kidneys, and spleen were aseptically collected, weighed, and homogenized; and the bacterial burdens were determined by CFU counts. Each symbol represents one animal ($N = 12/$ strain). Horizontal bars represent the mean from two pooled independent experiments. Significant differences were determined by one-way ANOVA with Dunnett's multiple-comparison test. *, $P < 0.05$ versus wt; ***, $P < 0.001$ versus wt; ****, $P < 0.0001$ versus wt.

among other unidentified iron acquisition systems, as none of the deletions created led to either complete depletion of the cell iron stores or significant changes in other parameters, such as bacterial survival in macrophages or human blood. This potential redundancy has been demonstrated in other microorganisms (30, 31), and our data are highly indicative that a similar process may also occur in *S. epidermidis*. After establishing the role of these loci in iron acquisition and taking into account our previous findings supporting the pivotal role of iron in *S. epidermidis* biofilm formation, we hypothesized that the lack of either *hts* or *fhuC* would be detrimental for biofilm formation. According to our results, *S. epidermidis* relies on iron acquisition through the Hts-FhuC transporter to form biofilms under iron-restricted conditions. The association between iron acquisition and biofilm formation has been established in some pathogens (32), but it is still poorly explored in *S. epidermidis*. Identifying new molecular factors involved in biofilm formation is key in the context of *S. epidermidis* infections, as biofilm formation is regarded as the major virulent trait exhibited by this pathogen.

Lastly, this study provides important *in vivo* data of the first iron acquisition mutants ever created in *S. epidermidis*. Although lack of either *hts* or *fhuC* did not significantly affect the bacterial survival within murine macrophage RAW264.7 cells or human macrophages, we demonstrated in a murine model of bacteremia that the Hts-FhuC iron

**FIG 3** Legend (Continued)

lysed macrophages, plated on TSA and CFU counted after 1 to 2 days at 37°C. (b) Gentamicin protection assays using murine RAW264.7 macrophages. Data are represented as mean ± SD of the fold change in CFU (relative to 0 h postinfection) for biological replicates ($N = 3$). (c) Gentamicin protection assays using hMDMs. Each point represents the CFU fold change (2 versus 0 h postinfection) obtained in one experiment with a single donor and lines represent the median from biological replicates ($N = 3$ donors). Values above and below 0 indicate bacterial replication and clearance, respectively. (b, c) Values above and below 0 indicate bacterial replication and clearance, respectively. No significant differences were detected using one-way ANOVA with Dunnett's multiple-comparison test. GM-CSF, granulocyte macrophage colony-stimulating factor; M-CSF, macrophage colony-stimulating factor.

transporter plays a significant role in bacterial survival *in vivo*. The absence of cytokine help may explain the inability of RAW264.7 cells alone to control *S. epidermidis* intracellular replication *in vitro* as interferon-γ was previously shown to limit *S. epidermidis* intracellular persistence in murine macrophages (25). Moreover, RAW264.7 cells do not express apoptotic speck-like protein with a caspase activation and recruiting domain and thus have no caspase-1 inflammasome activity, secreting lower mature interleukin 1β (IL-1β) levels (33). This is an important mechanism that participates in staphylococcal killing by macrophages (34). Additionally, our results indicate that M1-type polarization makes macrophages less permissive for *S. epidermidis* growth than M2-type polarization in line with previous observations with *S. aureus* (35). In the infected mouse groups, no major organ-dependent variation in the bacterial loads was observed for the mutants, suggesting that this iron acquisition mechanism does not play a niche-specific role, at least in the organs tested. The concept of iron acquisition-related molecules displaying a niche-specific role has been recently introduced by Perry et al., based on the observation that *S. aureus* siderophores were heterogeneously distributed across abscesses in different tissues (12). However, and as discussed above, *S. epidermidis* iron acquisition mechanisms are not as diverse as those expressed by *S. aureus*, and thus it seems they may play an equal role regardless of host location.

Collectively, this study represents the most thorough analysis to date of iron acquisition in *S. epidermidis*. The iron ABC transporter Hts-FhuC proved to be essential for bacterial survival in the iron-deprived environments, like those found within the host, since deletion of either *hts* or *fhuC* loci proved to be detrimental for two key virulence processes: biofilm formation and bacterial survival *in vivo*. Ultimately, this work emphasizes the potential of the Hts-FhuC iron transporter as a potential target for new therapeutic strategies against biofilm-associated *S. epidermidis* infections.

## MATERIALS AND METHODS

**Strains, plasmids, antibiotics, and culture media.** Bacterial strains and plasmids used in this study are described in Table S1. Unless otherwise noted, strains were cultured at 37°C. For genetic manipulations, *Escherichia coli* was grown in lysogeny broth (LB; 10 g/liter tryptone, 5 g/liter yeast extract, 10 g/liter NaCl). Staphylococci were grown in TSB (BD Diagnostic Systems, Heidelberg, Germany). Solid media were prepared by adding 1.5% (wt/vol) agar (BD) to the culture medium. For selection of plasmids and recombinant alleles, antibiotics (Sigma-Aldrich, St. Louis, MO) were added to the medium at the following concentrations: ampicillin (100 $\mu$g/mL) for *E. coli* selection and plasmid maintenance; trimethoprim (30 $\mu$g/mL), spectinomycin (150 $\mu$g/mL), and erythromycin (10 $\mu$g/mL) for staphylococci selection; and chloramphenicol (10 $\mu$g/mL) for staphylococcal plasmid maintenance. Iron restriction was achieved by slightly modifying a chemically defined medium (CDM) recipe (36), in which its original iron source (ammonium iron [II] sulfate) was omitted. This culture medium is henceforth referred to as CDM(Fe−). Iron-enriched conditions were achieved either by using TSB or by supplementing CDM(Fe−) with 10 $\mu$M FeCl$_3$ [CDM(Fe+)]. All solutions and media were made with water purified through a Milli-Q water purification system (Millipore, Burlington, MA).

**Genetic manipulations.** Standard DNA manipulations were performed essentially as described by Sambrook et al. (37). Restriction endonucleases were purchased from New England Biolabs, Inc. (Frankfurt, Germany) or Thermo Scientific Inc. (Waltham, MA). Phusion high-fidelity DNA polymerase and DyNAzyme II DNA polymerase were purchased from Thermo Scientific Inc. Plasmid DNA was purified using the QIAprep spin miniprep kit (Qiagen, Hilden, Germany) according to the manufacturer's instructions. For plasmid purification from staphylococci, the resuspension buffer provided with the plasmid isolation kit was supplemented with 25 U of lysostaphin (Sigma-Aldrich), and the cell suspension was incubated for 30 min at 37°C. Oligonucleotides and DNA sequencing services were purchased from Eurofins Genomics (Ebersberg, Germany).

**Construction of mutant strains.** An allelic replacement strategy was used for the construction of four deletion mutants in *S. epidermidis* 1457 strain. The list of primers used is shown in Table S2. For each mutant, two ~1-kb fragments flanking regions up- and downstream of the coding region to be deleted and an antibiotic resistance cassette were amplified using Phusion high-fidelity DNA polymerase (Thermo Fisher Scientific Inc.). Amplicons were ligated and cloned into plasmid pBASE6 (38) using (i) a Gibson assembly cloning kit (New England Biolabs, Inc.), according to the manufacturer's instructions or (ii) circular polymerase extension cloning (CPEC), as previously described (39). The resulting plasmids were introduced by electroporation, first into *S. aureus* RN4220 and then into *S. epidermidis* 1457Δ*agr* or 1457-M12. Next, using phage A6C, plasmids were introduced into *S. epidermidis* 1457. Selection of mutants was performed essentially as described (40). Correctness of the chromosomal mutations was verified using PCR with primers that bind to genetic regions not involved in the mutagenesis process, and afterwards the respective amplicons were sequenced. For complementation, DNA fragments containing the deleted coding sequences and their anticipated natural promoters were amplified and

cloned into plasmid pRB473, as described above. Plasmids were introduced by electroporation first into *S. aureus* PS187ΔΔ and then into *S. epidermidis* 1457 mutant strains using phage Φ187, following a previously published protocol (41). All wild-type and mutated alleles were confirmed for their sequence correctness by DNA sequencing.

**Quantification of bacterial iron content.** Two-mL of cultures grown overnight in TSB (BD) were harvested by centrifugation at $5,000 \times g$ for 10 min at 4℃. The cells were washed twice in ultrapure water and diluted into CDM(Fe−) to an optical density at 640 nm ($OD_{640}$) of 0.025 ($\sim 10^7$ CFU/mL) in disposable plastic tubes. Chloramphenicol was added to the growth medium of the plasmid-bearing strains for plasmid maintenance. Cultures were incubated at 37℃, 120 rpm (ES-20 Shaker-Incubator) for 24 h. Afterwards, the cultures were harvested by centrifugation at $5,000 \times g$, for 10 min at 4℃, and the pellet was washed thrice with metal-free ultrapure water to remove salts. Samples were then assayed for total iron content through atomic absorption spectrophotometry. Homogenized bacterial samples dispersed in ultrapure water were placed in previously weighed Teflon vessels (MS105, Mettler Toledo, Greifensee, Switzerland) and then dried in an oven at 90℃ (P Selecta, Barcelona, Spain) until three reproducible weight values were obtained. Microwave-assisted digestion of samples was performed by adding 10 mL of Suprapur nitric acid 65% (vol/vol) (Merck) to each vessel containing the dried and accurately weighed samples. The microwave-assisted digestion proceeded accordingly with the steps described in Table S3, using a Mars-X 1,500-W microwave accelerated reaction system for digestion and extraction (CEM Corp., Matthews, NC), configured with a 14-position carousel and equipped with pressure and temperature sensors. After digestion and cooling to approximately 30℃, the samples were kept frozen in polycarbonate containers at −20℃ until analysis. Lastly, iron quantification was carried out using an Analytik Jena ContrAA 700 high-resolution continuum source flame atomic absorption spectrometer (Analytik Jena, Jena, Germany) equipped with a xenon short-arc lamp XBO 301 (GLE, Berlin, Germany) with a nominal power of 300 W operating in a hot spot mode as a continuum radiation source. Iron was analyzed at 248.3270 nm by using the Graphite Furnace module equipped with an MPE60 autosampler (Analytik Jena) and argon 5.0 purity grade (Linde, Munich, Germany) as the inert gas. Transversal and pyrolytically coated graphite tubes with integrated platforms were used. In order to obtain maximum absorbance and minimum background values, operational parameters were optimized and are presented in Table S4. External calibration curves were daily constructed based on at least six standard solutions of iron prepared from 1,000 mg/liter stock solutions (Panreac Quimica SA, Barcelona, Spain). Magnesium nitrate hexahydrate (traceable to standard reference material [SRM] from National Institute of Standards and Technology [NIST]; Merck) was used as a matrix modifier at 0.1% (wt/vol). All glassware and plastic material were soaked in nitric acid (50% vol/vol), thoroughly rinsed with ultrapure water, and dried before use. The instrument performance was checked using analytical blanks and standards analyzed daily and regularly along with samples. The results were normalized to the cell dry weight.

**Detection of siderophore production.** Bacterial cultures were prepared as described above, except that the incubation period in CDM(Fe−) was 72 h for induction of maximal siderophore production. Afterwards, the cultures were harvested by centrifugation at $5,000 \times g$, for 10 min at 4℃. Culture supernatants were collected and filter-sterilized (pore size, 0.2 $\mu$m) for analysis of siderophore production using a modified Chrome Azurol S (CAS) agar diffusion assay as previously described (42).

**Planktonic growth curves.** Two-mL of cultures grown overnight in TSB were harvested by centrifugation at $5,000 \times g$ for 10 min. The cells were washed twice in 0.9% (wt/vol) NaCl and diluted into CDM (Fe−) to an $OD_{640}$ of 0.025 ($\sim 10^7$ CFU·mL$^{-1}$) in a conical glass flask. Flasks were incubated at 37℃, 120 rpm (ES-20 Shaker-Incubator). $OD_{640}$ was measured hourly up to 8 h and also at 24 h of incubation (when appropriate, concentrated samples were diluted in CDM(Fe−) for accurate measurement). Three independent experiments were performed for each condition tested.

**Biofilm formation assays.** Biofilms were grown either on 96-well microplates made of polystyrene plastic (Orange Scientific, Braine-l'Alleud, Belgium) for quantification of biofilm biomass or on Lab-Tek chamber slide system 8-well Permanox slides (Thermo Fisher Scientific Inc.) for confocal microscopy analysis. The cultures were prepared as described above, diluted into CDM(Fe−) to an $OD_{640}$ of 0.25 ($\sim 10^8$ CFU/mL), and further diluted 1:100 into (i) CDM(Fe−), (ii) CDM(Fe+), or (iii) TSB. Afterwards, diluted bacterial suspensions were placed into the microplates/chamber slides and incubated for 24 h at 37℃ under static conditions.

**Quantification of biofilm biomass.** Biofilms formed on 96-well microtiter plates were used for biomass quantification. After incubation, culture supernatants were removed carefully, and biofilms were washed twice with 200 $\mu$l of 0.9% NaCl and then stained by crystal violet technique, as previously described (43). Experiments were run at least in triplicate with technical quadruplicates for each condition tested.

**Confocal laser scanning microscopy analysis.** Biofilms formed on the chamber slide system were analyzed through confocal laser scanning microscopy (CLSM). After incubation, the culture supernatants were removed carefully, and the biofilms were washed twice with 200 $\mu$l of 0.9% NaCl and then stained with (i) DAPI (4′,6-diamidino-2-phenylindole) nucleic acid stain (Sigma-Aldrich) for visualization of cells and (ii) wheat germ agglutinin (WGA) conjugated with Texas Red (Thermo Fisher Scientific, Inc.) for staining of N-acetylglucosaminyl residues (PIA/PNAG). All staining procedures were performed according to the manufacturer's instructions. Stained biofilms were visualized as previously described (19).

**Gene expression analysis.** RNA samples were obtained from *S. epidermidis* cells cultured in CDM (Fe−) for 24 h. RNA extraction, DNase treatment and RNA quality determination were performed as previously described (19). cDNA synthesis was performed using the RevertAid first strand cDNA synthesis kit (Thermo Fisher Scientific, Inc.) following the manufacturer's instructions. The same amount of total RNA (300 ng) from each sample was reverse transcribed in a 10-$\mu$l reaction volume using random hexamer (or gene-specific) primers as priming strategy. To determine the possibility of genomic DNA

carryover, control reactions were performed under the same conditions but lacking the reverse transcriptase enzyme (NRT control). All RNA samples extracted were absent of significant genomic DNA. Gene expression was determined by qPCR using Xpert iFast SYBR Mastermix (GRiSP, Lda., Porto, Portugal). Each PCR contained 2 $\mu$l of 1:200 diluted cDNA or NRT control, 5 $\mu$l of master mix, 1 $\mu$l of primer mixture (in the final reaction, each primer was at 0.3 $\mu$M), and 2 $\mu$l of nuclease-free water. qPCR runs were performed on a CFX 96 (Bio-Rad) with the following cycle parameters: 95°C for 3 min and 40 cycles of 95°C for 5 s and 60°C for 25 s. Melt analysis was performed at the end to ensure the absence of unspecific products and primer dimer. All genes were quantified in duplicate for biological triplicates. The expression of the genes tested was normalized to the expression of the reference gene 16S rRNA. The data were log transformed (Log$_2$) before statistical analysis was performed. Information about the primers used in this study is listed in Table S2. Primers were designed with the aid of Primer3 (44) using *S. epidermidis* 1457 genome sequence (NCBI accession no. CP020463.1) as the template. mFold was used for prediction of secondary structures (45). No secondary structures were found for the operating temperatures used. Gene specificity of all primers was confirmed using Primer-BLAST (46). PCR amplification efficiency (E) for each gene has been previously determined (19).

**Isolation of peripheral blood mononuclear cells (PBMCs).** Human samples were obtained in agreement with the principles of the Declaration of Helsinki. PBMCs were isolated from surplus buffy coats, kindly provided by the Immunohemotherapy Department of Centro Hospitalar São João (CHSJ), Porto, Portugal. The procedures were approved by the Hospital Ethical Committee (protocol 90/19). Informed written consent that the by-products of their blood collections could be used for research purposes was obtained from the blood donors. The blood was diluted 1:2 in DPBS (Dulbecco's phosphate-buffered saline, without calcium and magnesium; Sigma-Aldrich), and 6 mL were carefully layered onto 3-mL Histopaque-1077 (Sigma-Aldrich) and centrifuged (400 $\times$ *g*, 30 min, room temperature, break-off) (Heraeus Megafuge 1.0R, Heraeus, Hanau, Germany). The PBMC layer was carefully transferred into a clean conical centrifuge tube and washed with DPBS. The cell pellet was recovered in DPBS, and cell concentration was determined.

**Monocyte purification by magnetic-activated cell sorting.** Monocytes were purified from previously prepared PBMC suspensions using CD14 MicroBeads, human kit (Miltenyi Biotec, Bergisch Gladbach, Germany), and MS columns (Miltenyi Biotec) in a Mini MACS separator (Miltenyi Biotec), according to manufacturer's instructions. The obtained CD14$^+$ cells were counted using a hemocytometer.

**Macrophage differentiation.** Monocytes (CD14$^+$ cells) were plated in either 6-, 24-, or 96-well cell culture plates (Nunclon Delta Surface; Thermo Fisher Scientific Inc.) in complete RPMI medium (cRPMI) (RPMI 1640 medium supplemented with 10 mM HEPES buffer, 2 mM L-glutamine, 100 U/mL penicillin/streptomycin, 0.05 mM $\beta$-mercaptoethanol, all from Sigma-Aldrich), 5% (vol/vol) heat-inactivated fetal bovine serum (FBS; Biowest, Riverside, MO) or autologous plasma (where indicated). The cells were seeded at the appropriate concentration and incubated at 37°C in a humidified atmosphere and 5% CO$_2$. To generate human monocyte-derived macrophages (hMDM) skewed toward an M1- or M2-like profile, cRPMI medium was supplemented with either 25 ng/mL of macrophage colony-stimulating factor (M-CSF; R&D Systems, Minneapolis, MN) or 25 ng/mL of granulocyte macrophage colony-stimulating factor (GM-CSF; PeproTech, Rocky Hill, NJ), respectively. Half of the culture medium was replaced at days 3 and 6 of culture by fresh, prewarmed M-CSF or GM-CSF supplemented cRPMI medium, as adequate. Experiments using hMDM were performed with cells prepared from three different donors.

**RAW264.7 cell cultures.** Murine RAW264.7 macrophages (ATCC TIB-71) were grown in cRPMI and incubated at 37°C in a humidified atmosphere and 5% CO$_2$. The used cells underwent less than 10 passages. After this point, new cells were revived from frozen stocks.

**Infection of macrophages: gentamicin protection assays.** Suspensions of *S. epidermidis* wt and its isogenic mutants were prepared as described above and used to infect previously plated RAW264.7 (5 $\times$ 10$^5$ cells/well) or hMDM (1 $\times$ 10$^5$ cells/well) in 96-well plates at an multiplicity of infection (MOI) of 10:1. To synchronize phagocytosis, the plates were centrifuged at 300 $\times$ *g* for 2 min followed by incubation at 37°C in the presence of 5% CO$_2$. Macrophages were allowed to internalize bacteria for 30 min. Afterwards, the culture supernatants were discarded, and prewarmed serum-free cRPMI plus 50 $\mu$g/mL gentamicin (AppliChem, Darmstadt, Germany) was added for 60 min to eliminate extracellular bacteria. After this treatment, macrophages were rinsed with DPBS and further incubated in prewarmed antibiotic-free cRPMI for the desired period of time (0, 2, 6, 12, and 24 h). Release of the gentamicin-protected bacteria (which corresponds to the intracellular fraction) was performed by lysing macrophages with 0.1% (wt/vol) saponin (Sigma-Aldrich) in PBS for 15 min. In order to eliminate bacterial aggregates, lysates underwent sonication (three cycles of 10 s at 30% amplitude using a Branson W140 Sonifier, Danbury, CT). Lastly, lysates were serially diluted in PBS and plated onto TSA plates for CFU enumeration.

**Survival in whole human blood.** Survival in whole human blood was carried out using an *ex vivo* human blood model of bloodstream infection (5). Human blood was collected in heparin tubes (Vacuette, GreinerBio-One GmbH, Austria) from seven healthy volunteers. The blood was collected under a protocol approved by the Institutional Review Board of the University of Minho (SECVS 002/2014 [ADENDA]), which is in strict accordance with the Declaration of Helsinki and Oviedo Convention. All donors gave written informed consent to have blood taken. Cultures were prepared as described above, diluted into CDM(Fe−) to an OD$_{640}$ of 0.25 ($\sim$10$^8$ CFU/mL), and further diluted in PBS to $\sim$10$^6$ CFU/mL. Finally, 50 $\mu$l of the bacterial suspension was mixed with 450 $\mu$l of blood in 2-mL microtubes. The mixture was incubated 0, 2, and 4 h at 37°C and 5% CO$_2$ with orbital rotation (80 rpm) and then serially diluted in PBS and plated onto TSA plates. Bacterial survival was assessed from the ratio between CFU counts at time points 2 and 4 h and those at 0 h. The data were log transformed (Log$_2$) before statistical analysis was performed (full details available in the supplemental material).

**Murine model of *S. epidermidis* bacteremia.** BALB/c mice were purchased from Charles River (Barcelona, Spain) and kept under specific pathogen-free conditions at the Animal Facility of the Instituto de Investigação e Inovação em Saúde (i3S; Porto, Portugal). Procedures involving mice were performed according to the European Convention for the Protection of Vertebrate Animals used for Experimental and Other Scientific Purposes (ETS 123), directive 2010/63/EU of the European parliament and of the council of 22 September 2010 on the protection of the animals used for scientific purposes, and Portuguese rules (DL 113/2013). The experiments were approved by the institutional board responsible for animal welfare (ORBEA) at i3S, and authorization to perform the experiments was issued by the competent national authority (Direcção Geral de Alimentação e Veterinária; reference number 014036/2019-07-24). *In vivo* infection experiments were performed following biosafety level 2 (BSL-2) guidelines. BALB/c mice were infected intravenously with ~$5 \times 10^8$ CFU of *S. epidermidis* 1457 or its isogenic siderophore biosynthetic ($\Delta sfa$) and iron uptake ($\Delta hts$ and $\Delta fhuC$) mutants. Six hours after infection, the mice were euthanized; blood, liver, kidneys, and spleen were aseptically collected, weighed, and homogenized; and bacterial burden were determined by plating for CFU counts on TSA. The data were obtained from two independent experiments.

**Sequence analysis.** DNA sequences were retrieved from the available genome of *S. epidermidis* 1457 (NCBI accession no. CP020463). Identification of putative Fur boxes was performed with the FIMO tool (47), using the default parameters and the 19-bp Fur box consensus sequence 5′-GATAATGATAATCATTATC-3′ (48) as the input motif. antiSMASH 5.1.0 was used for identification and analysis of secondary metabolite biosynthesis gene clusters (49). Synteny conservation was studied in all available *S. epidermidis* genomes using SyntTax (50).

**Statistical analysis.** Statistical significance was determined using GraphPad Prism version 7.0a. The statistical tests used, significance values, and group sizes are described in the figure legends. Significance was defined as $P < 0.05$, and data were excluded only on the basis of technical errors associated with the experiment.

## SUPPLEMENTAL MATERIAL

Supplemental material is available online only.

**SUPPLEMENTAL FILE 1**, PDF file, 2.8 MB.

## ACKNOWLEDGMENTS

We acknowledge the assistance of Nurse Filomena and thank all the blood donors.

N.C. and M.V. acknowledge the support by the Portuguese Foundation for Science and Technology (FCT) through the funded project PTDC/BIAMOL/29553/2017, under the scope of COMPETE2020 (POCI-01-0145-FEDER-029553). N.C. acknowledges the strategic funding of UID/BIO/04469/2019 unit by FCT. M.V. acknowledge the support of the i3S Scientific Platform HEMS, member of the Portuguese Platform of BioImaging (PPBI) (PPBIPOCI-01-0145-FEDER-022122), funded by FCT. H.R. acknowledges support from the German Research Council (DFG Ro 2413/4-1), the Damp Foundation (2013-19), and the Joachim Herz Foundation. F.O. was supported by an individual Ph.D. scholarship (SFRH/BD/101399/2014).

F.O., M.V., H.R., and N.C. designed research; F.O., T.L., A.M.S., and C.S. performed research; S.M. contributed new reagents/analytic tools; F.O., T.L., A.C., C.S., S.M., and S.W. analyzed data; M.V., H.R., and N.C. supervised research; and F.O., M.V., H.R., and N.C. wrote the paper. All authors read and approved the final version of the manuscript.

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
