## [Reviewer comments · Microbiology Spectrum]

Microbiology Spectrum

Involvement of the iron-regulated loci *hts* and *fhuC* in biofilm formation and survival of *Staphylococcus epidermidis* within the host

Fernando Oliveira, Tânia Lima, Alexandra Correia, Ana Silva, Cristina Soares, Simone Morais, Samira Weißelberg, Manuel Vilanova, Holger Rohde, and Nuno Cerca

Corresponding Author(s): Nuno Cerca, University of Minho

Review Timeline:

Submission Date:	November 5, 2021
Editorial Decision:	November 23, 2021
Revision Received:	December 2, 2021
Accepted:	December 5, 2021

Editor: Hermine Mkrtchyan

Reviewer(s): The reviewers have opted to remain anonymous.

Transaction Report:

DOI: <https://doi.org/10.1128/spectrum.02168-21>

November 23, 2021

Dr. Nuno Cerca
University of Minho
Campus de Gualtar
Braga
Portugal

Re: Spectrum02168-21 (Involvement of the iron-regulated loci *hts* and *fhuC* in biofilm formation and survival of *Staphylococcus epidermidis* within the host)

Dear Dr. Nuno Cerca:

Thank you for submitting your manuscript to Microbiology Spectrum. Your manuscript has been reviewed by two experts in the field. I invite you to resubmit your manuscript after addressing the reviewers comments below, including the experimental suggestions. Please, note that abbreviations such as TSB, TSA must be written in full for the first time.

Link Not Available

Sincerely,

Hermine Mkrtchyan

Journals Department
Reviewer comments:

Reviewer #1 (Comments for the Author):

In the manuscript, "Involvement of the iron-regulated loci *hts* and *fhuC* in biofilm formation and survival of *Staphylococcus epidermidis* within the host" by Oliveira, F. et al., the authors describe the role of the *hts* and *fhuC* genes in iron acquisition as well as their role in *S. epidermidis* pathogenesis. The authors demonstrate that these genes are conserved across *S. epidermidis* and that deletion of these genes delays growth and decreases biofilm formation in iron-limiting conditions. While these genes do not play a role in *S. epidermidis* survival when co-cultured with human and murine macrophages or in human whole blood, the *S. epidermidis* strains with each of these genes deleted are attenuated for pathogenesis in a murine systemic infection model. The manuscript is well written, and the data are generally support the conclusions, but this reviewer has several comments that need to be addressed and has a few experimental requests.

1. On line 60, this reviewer thinks that "this" should be replaced with "these".

2. The concluding sentence starting on line 78 is a bit confusing to this reviewer. Please attempt to rewrite this sentence for increased clarity.
3. The axis of Figure 1c says Mn, but the text and figure legend says Fe.
4. The text claims that Fe is increased in the hts mutant (Figure 1c), however, there are no statistics on the graph. If the hts strain is not statistically significant compared to wt, the claims of increased Fe cannot be made in the text. If the Fe increase is statistically significant, please add some speculation as to what this means.
5. The 24 h time point used to measure intracellular Fe is a late time point for metal uptake assays. Please include earlier time points, preferably consistent with the growth defects observed in Figure 1d, in this analysis because as shown in Figure 1d, all of the strains reach a similar OD by 24 h.
6. Please repeat the intracellular Fe experiment with the complementation strains shown in Figure 2 to demonstrate that the Fe differences are directly a result of the deleted genes.
7. Please include growth curve data in Figure 1d for CDM (Fe+) to determine if Fe addition complements the observed growth phenotypes.
8. Please include growth curve data in Figure 1d for the genetically complemented strains shown in Figure 2.
9. This reviewer finds the concluding sentence starting on line 107 confusing. Please attempt to reword this sentence for clarity.
10. The text on line 140 claims that the bacteria are replicating in RAW264.7 macrophages as seen in Figure 3b. However, no statistics are noted on the graphs, and the figure legend says explicitly that there were no significant differences. Therefore, these data do not support the assertion that *S. epidermidis* is replicating in RAW264.7 macrophages, and these claims need to be amended or removed.
11. In Figure 3c, please show all of the time points as in Figure 3b, and remove "data not shown" from line 147.
12. Line 150 calls out to figure 3d, which is not present in the manuscript.
13. Figure 4 can be moved into supplemental.

Reviewer #2 (Comments for the Author):

The study investigated the role of the gene loci *hts* and *fhu* in iron acquisition in *S. epidermidis*.

Although the manuscript is relatively well written, the scientific part is somewhat descriptive and fragmented. There was no red thread that goes through the study as explained below.

-The loci were speculated to be involved in iron acquisition. When these loci were deleted the resulting cells behaved differently. A reduction in the iron cell content was found in $\Delta fhuC$ cells while

an increased iron cell content was surprisingly found in Δhts cells. This is a very interesting result but the authors did not make any attempt to pursue further to explain the molecular mechanisms behind this contrasting regulation pattern.

-qPCR was performed and the gene *sfaB* was found to be upregulated in the deletion strains. However, no further experiment was performed to connect the biological aspect of this gene in this context.

-The biofilm ability of both mutants was impaired under iron-limited but not under iron-rich conditions. Any molecular explanation for this? Is iron important for biofilm formation, and how?

-The intracellular growth (??) of the mutants was not affected in the murine RAW264.7 macrophages but it was affected in human macrophages hMDMs. No further experiment was conducted to explain this contrasting result.

-The mutants were not affected in human blood but were affected in a murine in vivo model of *S. epidermidis* bacteremia. Why and how?

Staff Comments:

Preparing Revision Guidelines

To submit your modified manuscript, log onto the eJP submission site at <https://spectrum.msubmit.net/cgi-bin/main.plex>. Go to Author Tasks and click the appropriate manuscript title to begin the revision process. The information that you entered when you

first submitted the paper will be displayed. Please update the information as necessary. Here are a few examples of required updates that authors must address:

Please return the manuscript within 60 days; if you cannot complete the modification within this time period, please contact me. If you do not wish to modify the manuscript and prefer to submit it to another journal, please notify me of your decision immediately so that the manuscript may be formally withdrawn from consideration by Microbiology Spectrum.

In the manuscript, "Involvement of the iron-regulated loci *hts* and *fhuC* in biofilm formation and survival of *Staphylococcus epidermidis* within the host" by Oliveira, F. et al., the authors describe the role of the *hts* and *fhuC* genes in iron acquisition as well as their role in *S. epidermidis* pathogenesis. The authors demonstrate that these genes are conserved across *S. epidermidis* and that deletion of these genes delays growth and decreases biofilm formation in iron-limiting conditions. While these genes do not play a role in *S. epidermidis* survival when co-cultured with human and murine macrophages or in human whole blood, the *S. epidermidis* strains with each of these genes deleted are attenuated for pathogenesis in a murine systemic infection model. The manuscript is well written, and the data are generally support the conclusions, but this reviewer has several comments that need to be addressed and has a few experimental requests.

1. On line 60, this reviewer thinks that "this" should be replaced with "these".
2. The concluding sentence starting on line 78 is a bit confusing to this reviewer. Please attempt to rewrite this sentence for increased clarity.
3. The axis of Figure 1c says Mn, but the text and figure legend says Fe.
4. The text claims that Fe is increased in the Δ *hts* mutant (Figure 1c), however, there are no statistics on the graph. If the Δ *hts* strain is not statistically significant compared to wt, the claims of increased Fe cannot be made in the text. If the Fe increase is statistically significant, please add some speculation as to what this means.
5. The 24 h time point used to measure intracellular Fe is a late time point for metal uptake assays. Please include earlier time points, preferably consistent with the growth defects observed in Figure 1d, in this analysis because as shown in Figure 1d, all of the strains reach a similar OD by 24 h.
6. Please repeat the intracellular Fe experiment with the complementation strains shown in Figure 2 to demonstrate that the Fe differences are directly a result of the deleted genes.
7. Please include growth curve data in Figure 1d for CDM (Fe⁺) to determine if Fe addition complements the observed growth phenotypes.
8. Please include growth curve data in Figure 1d for the genetically complemented strains shown in Figure 2.
9. This reviewer finds the concluding sentence starting on line 107 confusing. Please attempt to reword this sentence for clarity.
10. The text on line 140 claims that the bacteria are replicating in RAW264.7 macrophages as seen in Figure 3b. However, no statistics are noted on the graphs, and the figure legend says explicitly that there were no significant differences. Therefore, these data do not support the assertion that *S. epidermidis* is replicating in RAW264.7 macrophages, and these claims need to be amended or removed.
11. In Figure 3c, please show all of the time points as in Figure 3b, and remove "data not shown" from line 147.
12. Line 150 calls out to figure 3d, which is not present in the manuscript.
13. Figure 4 can be moved into supplemental.

Authors: We thank the reviewers for the critical and constructive comments, as well for the timely review. On the revised manuscript we have included some new experimental results, and also improved upon the writing of the manuscript. Bellow, we provide a point-by-point answer to all the comments raised by the reviewers.

We would kindly request a quick re-revision of our manuscript, due to time constraints related to our research project coming to and end, that affects our ability to pay for APC charges.

Reviewer #1 (Comments for the Author):

In the manuscript, "Involvement of the iron-regulated loci *hts* and *fhuC* in biofilm formation and survival of *Staphylococcus epidermidis* within the host" by Oliveira, F. et al., the authors describe the role of the *hts* and *fhuC* genes in iron acquisition as well as their role in *S. epidermidis* pathogenesis. The authors demonstrate that these genes are conserved across *S. epidermidis* and that deletion of these genes delays growth and decreases biofilm formation in iron-limiting conditions. While these genes do not play a role in *S. epidermidis* survival when co-cultured with human and murine macrophages or in human whole blood, the *S. epidermidis* strains with each of these genes deleted are attenuated for pathogenesis in a murine systemic infection model. The manuscript is well written, and the data are generally support the conclusions, but this reviewer has several comments that need to be addressed and has a few experimental requests.

1. On line 60, this reviewer thinks that "this" should be replaced with "these".

Answer(A): We have followed the reviewer's suggestion and change the word accordingly.

2. The concluding sentence starting on line 78 is a bit confusing to this reviewer. Please attempt to rewrite this sentence for increased clarity.

A: We appreciate the reviewer's suggestion to improve the clarity of our manuscript. We have slightly adjusted that sentence and hope it is now clearer (lines 78-79).

3. The axis of Figure 1c says Mn, but the text and figure legend says Fe.

A: We do apologize for that mistake. We have corrected the yy axis of Figure 1c.

4. The text claims that Fe is increased in the *hts* mutant (Figure 1c), however, there are no statistics on the graph. If the *hts* strain is not statistically significant compared to wt, the claims of increased Fe cannot be made in the text. If the Fe increase is statistically significant, please add some speculation as to what this means.

A: Although the difference between the *hts* mutant and wt is not statistically significant, it was observed an increase in the mean iron content of this mutant, which was partly reversed in its corresponding complementation strain (*phts*) (updated Figure 1c). In that sense, we consider this finding is worth mentioning. Nevertheless, and into account the reviewer's feedback, we have slightly adjusted that sentence (line 96).

5. The 24 h time point used to measure intracellular Fe is a late time point for metal uptake assays. Please include earlier time points, preferably consistent with the growth defects observed in Figure 1d, in this analysis because as shown in Figure 1d, all of the strains reach a similar OD by 24 h.

A: We understand the reviewer suggestion, and in fact we initially aimed to quantify

iron intracellular both at log phase and stationary phase. However, at 6-8h of growth, both mutants had a significant lower bacterial density, and we were not able, under our experimental conditions, to achieve the bacterial concentration required for an accurate, reliable quantification of the intracellular iron. Most importantly, although all strains had reached similar OD's at the stationary phase, we were still able to detect important differences in their intracellular iron content, which holds particularly true for the *fhuC* mutant. Taken together, although the reviewer's reasoning for testing earlier time points is valid, and would increase the depth of our experiments and the paper impact, we must argue that our experimental approach allowed us to confirm our hypothesis, which was the involvement of these two loci, *hts* and *fhuC*, in iron acquisition.

6. Please repeat the intracellular Fe experiment with the complementation strains shown in Figure 2 to demonstrate that the Fe differences are directly a result of the deleted genes.

A: The reviewer is absolutely right regarding this matter. Of course, we measured the iron cell content of the complementation strains in parallel with our wt and mutant strains. We do apologize for not having included this data in our initial submission, which is now shown in the revised version (Figure S2a) and mentioned in the main text (lines 109-112). Of note, on the revised submission we had included a new figure that has all the complementation experiments performed, not only the ones requested by this reviewer.

7. Please include growth curve data in Figure 1d for CDM (Fe+) to determine if Fe addition complements the observed growth phenotypes.

A: We understand the reviewer suggestion and we do agree that by performing this experiment we could provide even more information about the characterization of these mutants. However, the experiments suggested are not fundamental to the story we are portraying here.

We would like the reviewer to consider two aspects. First, it is this journal editorial guidelines to evaluate manuscripts solely on the basis of technical soundness regardless of potential impact. Therefore, while the requested experiment would increase the depth and impact of the paper, it should not be seen as a required experiment needed for this paper to be published. Secondly, while normally we would be interested in performing what experiments required in order to enhance the quality of the paper (regardless of journal editorial policies), unfortunately we will not be able to perform any more experiments, since we no longer have funds, human resources or time allocated to these projects in order to perform additional experiments.

8. Please include growth curve data in Figure 1d for the genetically complemented strains shown in Figure 2.

A: As mentioned on question 6, we have now included all the complementation experiments we have performed. In an attempt to complement the *fhuC* mutant, *in trans* expression from its natural promoter did not restore the growth phenotype exhibited by the mutant. In our experience, this might be associated to relevant gene doses effects when the gene is expressed from a multi copy plasmid (we used pRB473). In fact, we performed a comparative qPCR analysis of *fhuC* expression and observed that *fhuC* expression was ~12-fold higher in the complemented strain (*pfhuC*) vs wild type (wt). We apologize for not having included this data in our initial submission, which is now shown in the revised version (Figure S2b) and mentioned in the main text (lines 109-112).

9. This reviewer finds the concluding sentence starting on line 107 confusing. Please attempt to reword this sentence for clarity.

A: We have reworded that sentence (lines 111-113).

10. The text on line 140 claims that the bacteria are replicating in RAW264.7 macrophages as seen in Figure 3b. However, no statistics are noted on the graphs, and the figure legend says explicitly that there were no significant differences. Therefore, these data do not support the assertion that *S. epidermidis* is replicating in RAW264.7 macrophages, and these claims need to be amended or removed.

A: There was some absolute variation between experiment, but the trend was always the same, with an increase in bacterial concentration at 24 h. So, while the averaged values will lack statistical significance, we believe we should point out this observed phenomenon. However, we agree with the reviewer that we should be more conservative, and as such we have changed that sentence accordingly (lines 150-153).

11. In Figure 3c, please show all of the time points as in Figure 3b, and remove "data not shown" from line 147.

A: As stated in our manuscript, no CFUs were recovered at 24 h post-infection. Since we decided to represent the data as a log fold change in CFU (relative to 0 h post-infection), it is not possible (mathematically speaking) to perform that calculation for that time point. An alternative would be to represent our data in absolute CFU values, although we believe our initial approach is more intuitive. Nevertheless, we reworded that sentence and removed the "data not shown" part as it is redundant.

12. Line 150 calls out to figure 3d, which is not present in the manuscript.

A: There is no panel 3d and we have corrected the main manuscript. It was a typo and we do apologize for that.

13. Figure 4 can be moved into supplemental.

A: We have followed the reviewer's suggestion.

Reviewer #2 (Comments for the Author):

The study investigated the role of the gene loci *hts* and *fhu* in iron acquisition in *S. epidermidis*.

Although the manuscript is relatively well written, the scientific part is somewhat descriptive and fragmented. There was no red thread that goes through the study as explained below.

-The loci were speculated to be involved in iron acquisition. When these loci were deleted the resulting cells behaved differently.

A reduction in the iron cell content was found in $\Delta fhuC$ cells while an increased iron cell content was surprisingly found in Δhts cells. This is a very interesting result but the authors did not make any attempt to pursue further to explain the molecular mechanisms behind this contrasting regulation pattern.

A: We are afraid the reviewer missed some important parts of our manuscript. We do agree this was an interesting finding and we were really interested in understanding the mechanisms behind this phenomenon. We performed additional experiments (qPCR) in an attempt to understand the transcription levels of other iron acquisition-related genes in each of the mutant backgrounds. The results are presented throughout lines 101-116 and discussed throughout lines 187-194.

Of course, we understand part of the reviewer's feedback and we do agree that additional work would be required to better understand the observed phenomenon. Still, that is probably the case with most research papers: each experiment yields some results that in turn bring more questions and the need to perform an endless number of additional experiments. The further a research study goes, the higher the impact that study will likely have. However, we would like the reviewer to consider the editorial guidelines of Microbiology Spectrum, wherein manuscripts should be evaluated solely on the basis of technical soundness regardless of potential impact. Therefore, the fact that our experimental results are partly descriptive should not be considered as a flaw. We do hope the reviewer is sensible to the fact that we will not be able to perform more experiments that would complement the included experiments, since the research project will come to end in December, and we no longer have staff, time, and funds to perform additional experiments. Most importantly, we believe our manuscript brings novel information to the field, and that our data is scientific sound.

qPCR was performed and the gene *sfaB* was found to be upregulated in the deletion strains. However, no further experiment was performed to connect the biological aspect of this gene in this context.

A: We apologize for not being clear why we decided to include this gene. On the revised version of the manuscript we now have clarified our rationale (lines 105-106). This gene (*sfaB*) belongs to an iron-regulated locus (*sfaABCD*) that we recently characterized and demonstrated to play an important role in siderophore production (<https://www.frontiersin.org/articles/10.3389/fmed.2021.799227/abstract>), and for that reason, we wanted to test if any of the 2 newly described mutants could somehow directly affect the *sfa* operon.

The biofilm ability of both mutants was impaired under iron-limited but not under iron-rich conditions. Any molecular explanation for this? Is iron important for biofilm formation, and how?

A: We have previously demonstrated that *S. epidermidis* is largely dependent on iron availability to form biofilms (Oliveira, et al. Int J Med Microbiol. 2017 Dec;307(8):552-563. doi: 10.1016/j.ijmm.2017.08.009), which paved the way for the work we are now presenting in this manuscript. Unfortunately, we do not have a molecular explanation for that, other than the general iron requirement for the different biochemical processes involved in bacterial proliferation and biofilm formation.

The intracellular growth (??) of the mutants was not affected in the murine RAW264.7 macrophages but it was affected in human macrophages hMDMs. No further experiment was conducted to explain this contrasting result.

A: While the reviewer is right that we did not further explore this difference, we did provide a possible justification for these results. Importantly, it is well known that different experimental models can yield different and contrasting results but trying to determine the difference between RAW264.7 and primary human macrophages would be a whole new project.

On our original submission (lines 215-217), we wrote that "*The absence of cytokine help may explain the inability of RAW264.7 cells alone to control S. epidermidis intracellular replication in vitro as interferon- γ was previously shown to limit S. epidermidis intracellular persistence in murine macrophages*". On the revised version (lines 217-220), we added the following sentences and references: "Moreover, RAW264.7 cells do not express apoptotic speck-like protein with a caspase activation and recruiting domain and thus have no caspase-1 inflammasome activity, secreting lower mature IL-1- β levels (Pelegri et al., 2008). This is an important mechanism that participates in staphylococcal killing by macrophages (Shimada et al., 2010)".

We hope the reviewer acknowledges that further experiments are out of the scope of this manuscript, and that our experiments and conclusions follow the publishing editorial guidelines of Microbiology Spectrum.

The mutants were not affected in human blood but were affected in a murine in vivo model of *S. epidermidis* bacteremia. Why and how?

A: Like the previous question, herein we are addressing two very different experimental models, in order to give a wider view of our research question. Differences between two very distinct models can be expected. The *ex vivo* whole human blood infection model mostly relies on the activity of neutrophils that readily associate with staphylococcal cells, as previously shown in this model for *S. aureus* (Lehnert et al., 2021). The fast clearance of the bacteria might not allow putative bacterial recognition alterations resulting from the mutations under study to be highlighted. Moreover, *S. epidermidis* mechanisms counteracting neutrophil-mediated killing are less effective than those displayed by *S. aureus* (Otto, 2014). Besides different effector effectiveness of human vs murine immune cells facing staphylococci infections can always be considered. The more complex *in vivo* model involves several other factors that may affect infection outcome. In fact, attenuation of staphylococcal mutants highlighted using *in vivo* murine models may not translate into observable differences in survival using in whole blood model of human and mouse origin (Das et al., 2016). Endothelial adhesion and endothelial barrier disruption is an important event in disseminated bacterial infections (Lemichez et al., 2010). Lastly, iron regulation has been shown to impact adhesion of *S. aureus* to endothelial cells (Alfeo et al., 2021), and therefore one might assume that our *S. epidermidis* mutants may also present a disadvantage in that regard that would only be stressed using the *in vivo* infection model.

Overall, there is strong evidence available supporting the differences we observed between the *ex vivo* and *in vivo* models. Nevertheless, it is our understanding that this discussion should not be part of the main text, unless the reviewer and/or editor considers it absolutely necessary.

References

- Alfeo, M. J., Pagotto, A., Barbieri, G., Foster, T. J., Vanhoorelbeke, K., De Filippis, V., et al. (2021). Staphylococcus aureus iron-regulated surface determinant B (IsdB) protein interacts with von Willebrand factor and promotes adherence to endothelial cells. *Sci. Rep.* 11, 22799. doi:10.1038/S41598-021-02065-W.
- Das, S., Lindemann, C., Young, B. C., Muller, J., Österreich, B., Ternet, N., et al. (2016). Natural mutations in a Staphylococcus aureus virulence regulator attenuate cytotoxicity but permit bacteremia and abscess formation. *Proc. Natl. Acad. Sci. U. S. A.* 113, E3101–E3110. doi:10.1073/PNAS.1520255113.
- Lehnert, T., Leonhardt, I., Timme, S., Thomas-Rüddel, D., Bloos, F., Sponholz, C., et al. (2021). Ex vivo immune profiling in patient blood enables quantification of innate immune effector functions. *Sci. Rep.* 11. doi:10.1038/S41598-021-91362-5.
- Lemichez, E., Lecuit, M., Nassif, X., and Bourdoulous, S. (2010). Breaking the wall: targeting of the endothelium by pathogenic bacteria. *Nat. Rev. Microbiol.* 8, 93–104. doi:10.1038/NRMICRO2269.
- Otto, M. (2014). Staphylococcus epidermidis pathogenesis. *Methods Mol. Biol.* 1106, 17–31. doi:10.1007/978-1-62703-736-5_2.
- Pelegri, P., Barroso-Gutierrez, C., and Surprenant, A. (2008). P2X7 receptor differentially couples to distinct release pathways for IL-1beta in mouse macrophage. *J. Immunol.* 180, 7147–7157. doi:10.4049/JIMMUNOL.180.11.7147.
- Shimada, T., Park, B. G., Wolf, A. J., Brikos, C., Goodridge, H. S., Becker, C. A., et al. (2010). Staphylococcus aureus evades lysozyme-based peptidoglycan digestion that links phagocytosis, inflammasome activation, and IL-1beta secretion. *Cell*

Host Microbe 7, 38–49. doi:10.1016/J.CHOM.2009.12.008.

December 5, 2021

Dr. Nuno Cerca
University of Minho
Campus de Gualtar
Braga
Portugal

Re: Spectrum02168-21R1 (Involvement of the iron-regulated loci hts and fhuC in biofilm formation and survival of Staphylococcus epidermidis within the host)

Dear Dr. Nuno Cerca:

Your manuscript has been accepted, and I am forwarding it to the ASM Journals Department for publication. You will be notified when your proofs are ready to be viewed.

Sincerely,

Hermine Mkrtchyan
Editor, Microbiology Spectrum
